# Polybrominated Diphenyl Ethers (PBDEs) and Human Health: Effects on Metabolism, Diabetes and Cancer

**DOI:** 10.3390/cancers15174237

**Published:** 2023-08-24

**Authors:** Valerio Renzelli, Marco Gallo, Lelio Morviducci, Giampiero Marino, Alberto Ragni, Enzo Tuveri, Antongiulio Faggiano, Rossella Mazzilli, Annalisa Natalicchio, Maria Chiara Zatelli, Monica Montagnani, Stefano Fogli, Dario Giuffrida, Antonella Argentiero, Romano Danesi, Stella D’Oronzo, Stefania Gori, Tindara Franchina, Antonio Russo, Matteo Monami, Laura Sciacca, Saverio Cinieri, Annamaria Colao, Angelo Avogaro, Graziano Di Cianni, Francesco Giorgino, Nicola Silvestris

**Affiliations:** 1Italian Association of Clinical Diabetologists, 00192 Rome, Italy; valerio.renzelli@gmail.com; 2Endocrinology and Metabolic Diseases Unit, AO SS Antonio e Biagio e Cesare Arrigo of Alessandria, 15121 Alessandria, Italy; alberto.ragni@ospedale.al.it; 3Diabetology and Nutrition Unit, Department of Medical Specialities, ASL Roma 1, S. Spirito Hospital, 00193 Rome, Italy; leliomorviducci@gmail.com; 4Internal Medicine Department, Ospedale dei Castelli, Asl Roma 6, 00040 Ariccia, Italy; giampiero.marino@aslroma6.it; 5Diabetology, Endocrinology and Metabolic Diseases Service, ASL-Sulcis, 09016 Iglesias, Italy; enzo.tuveri@gmail.com; 6Endocrinology Unit, Department of Clinical & Molecular Medicine, Sant’Andrea Hospital, Sapienza University of Rome, 00189 Rome, Italy; antongiulio.faggiano@uniroma1.it (A.F.); rossella.mazzilli@uniroma1.it (R.M.); 7Department of Precision and Regenerative Medicine and Ionian Area, Section of Internal Medicine, Endocrinology, Andrology and Metabolic Diseases, University of Bari Aldo Moro, 70121 Bari, Italy; annalisa.natalicchio@uniba.it (A.N.); francesco.giorgino@uniba.it (F.G.); 8Section of Endocrinology, Geriatrics and Internal Medicine, Department of Medical Sciences, University of Ferrara, 44121 Ferrara, Italy; ztlmch@unife.it; 9Department of Precision and Regenerative Medicine and Ionian Area, Section of Pharmacology, University of Bari Aldo Moro, 70121 Bari, Italy; monica.montagnani@uniba.it; 10Clinical Pharmacology and Pharmacogenetics Unit, Department of Clinical and Experimental Medicine, University of Pisa, 56126 Pisa, Italy; stefano.fogli@unipi.it (S.F.); romano.danesi@unipi.it (R.D.); 11Department of Oncology, Istituto Oncologico del Mediterraneo, Viagrande, 95029 Catania, Italy; dgiuff57@gmail.com; 12Medical Oncology Unit, IRCCS Istituto Tumori “Giovanni Paolo II”, 70124 Bari, Italy; argentieroantonella@gmail.com; 13Interdisciplinary Department of Medicine, University of Bari Aldo Moro, 70121 Bari, Italy; stella.doronzo@uniba.it; 14Oncologia Medica, IRCCS Ospedale Don Calabria-Sacro Cuore di Negrar, 37024 Verona, Italy; stefania.gori@sacrocuore.it; 15Medical Oncology Unit, Department of Human Pathology “G. Barresi”, University of Messina, 98122 Messina, Italy; tindifra@yahoo.it (T.F.); nicola.silvestris@unime.it (N.S.); 16Department of Surgical, Oncological and Oral Sciences, Section of Medical Oncology, University of Palermo, 90133 Palermo, Italy; antonio.russo@usa.net; 17Diabetology, Careggi University Hospital, University of Florence, 50134 Florence, Italy; matteo.monami@unifi.it; 18Department of Clinical and Experimental Medicine, Endocrinology Section, University of Catania, 95124 Catania, Italy; laura.sciacca@unict.it; 19Medical Oncology Division, Breast Unit, Senatore Antonio Perrino Hospital, ASL Brindisi, 72100 Brindisi, Italy; saverio.cinieri@me.com; 20Endocrinology, Diabetology and Andrology Unit, Department of Clinical Medicine and Surgery, Federico II University of Naples, 80138 Naples, Italy; colao@unina.it; 21UNESCO Chair, Education for Health and Sustainable Development, Federico II University, 80131 Naples, Italy; 22Department of Medicine, Section of Diabetes and Metabolic Diseases, University of Padova, 35122 Padova, Italy; angelo.avogaro@unipd.it; 23Diabetes Unit, Livorno Hospital, 57100 Livorno, Italy; graziano.dicianni@uslnordovest.toscana.it

**Keywords:** PBDE, endocrine disruptors, environment, metabolism, diabetes, obesity, cancer

## Abstract

**Simple Summary:**

Endocrine disruptors are chemicals that can interfere and interact with the endocrine system, resulting in altered hormonal signaling and function. PBDEs are common endocrine disruptors that have been commonly used in industrial products, and their environmental accumulation has become a rising concern. Human exposure to PBDEs has been shown to influence glucose metabolism, thyroid and ovarian function and potentially affect cancer risk. Evidence, however, is often conflicting. This narrative review provides a comprehensive overview of the potential role of PBDEs in human health, with a particular focus on glucose metabolism, endocrine diseases and cancer. A deeper understanding of the complex interplay between exposure to endocrine disruptors, on one side, and obesity, diabetes, related metabolic disturbances and cancer, on the other side, can help guide public health interventions, in order to reduce the burden of these major social threats.

**Abstract:**

There is increasing evidence of the role of endocrine disruptors (EDs) derived from commonly employed compounds for manufacturing and processing in altering hormonal signaling and function. Due to their prolonged half-life and persistence, EDs can usually be found not only in industrial products but also in households and in the environment, creating the premises for long-lasting exposure. Polybrominated diphenyl ethers (PBDEs) are common EDs used in industrial products such as flame retardants, and recent studies are increasingly showing that they may interfere with both metabolic and oncogenic pathways. In this article, a multidisciplinary panel of experts of the Italian Association of Medical Diabetologists (AMD), the Italian Society of Diabetology (SID), the Italian Association of Medical Oncology (AIOM), the Italian Society of Endocrinology (SIE) and the Italian Society of Pharmacology (SIF) provides a review on the potential role of PBDEs in human health and disease, exploring both molecular and clinical aspects and focusing on metabolic and oncogenic pathways.

## 1. Introduction

Endocrine disruptors (EDs) are chemicals that can interfere and interact with the endocrine system, resulting in altered hormonal signaling and function. EDs can usually be found in industrial as well as in household products, being commonly employed for manufacturing and processing. In recent years, environmental accumulation has become a rising concern, due to most EDs having prolonged half-life and persistence [1].

Polybrominated diphenyl ethers (PBDEs) are a type of ED that have been commonly employed in a lot of industrial equipment and goods as flame retardants (see Table 1). Due to their physicochemical properties, they have been classified as persistent organic pollutants by the 2009 and 2017 Stockholm Conventions [2]. Therefore, uses of some commercial PBDE mixtures have been restricted [3,4]. However, high levels of PBDEs have recently been measured in cetaceans and endangered species [5].

PBDEs are among customary types of flame retardants detected near electronic garbage disposal recycling sites [6]. Despite regulations, the rapid turnover of electronic devices has led to an increasing amount of e-waste with rising concerns for environmental contamination.

A multidisciplinary panel of experts from five Italian multidisciplinary scientific societies (The Italian Association of Medical Diabetologists (AMD), the Italian Society of Diabetology (SID), the Italian Association of Medical Oncology (AIOM), the Italian Society of Endocrinology (SIE) and the Italian Society of Pharmacology (SIF)) met to review available evidence concerning the effects of PBDEs on many aspects of human health, with a particular focus on glucose metabolism, endocrine diseases and cancer. Different areas of expertise of each society aided in analyzing the various aims of this review, such as PBDEs’ toxicokinetics (TK) and toxicodynamics (TD), as well as molecular, metabolic and clinical effects of human exposure.
cancers-15-04237-t001_Table 1Table 1Industrial and commercial products in which PBDEs are commonly found (Modified from [7]).ApplicationsArticlesElectronic equipment and wastesCircuit boardsProtective coatings and casings (PCs, TVs, office equipment)Cable and wire sheetsFurniture and textilesCushioning and upholstery materialsCarpet coatingsPaintsTransportationCushioning and upholstery materialsAutomotive seatingConstruction materialsSound insulationPackagingRigid polyurethane foam constructionDrillingDrilling oils


## 2. Toxicokinetics and Toxicodynamics

From a chemical point of view, PBDEs are characterized by a common diphenyl ether molecule in which any of the 10 hydrogen atoms can be exchanged with bromine atoms [8]. The entire family of PBDEs consists of 209 congeners that, rather than single chemical compounds, are mixtures of several brominated entities. As flame-retardant additives, they are physically mixed—not chemically bound—into product applications and therefore easily released from the plastic matrix and able to migrate into the environment [9]. Despite the increasing concerns about risk effects resulting from prolonged exposure to PBDEs, several aspects of their TK and TD profile are still incompletely known.

### 2.1. Routes of Exposure

In large part, information related to the consequences of PBDE exposure for health originates from studies on laboratory animals orally administered with PBDEs and from human epidemiological data where the main exposure route is undefined but predictably oral [9]. The route of exposure to specific PBDE congeners may vary according to their nature and persistence in the environment (see Figure 1). Among the three PBDE mixtures more frequently used in industry (i.e., penta-, octa- and decaBDEs), decaBDEs have been often employed in television cabinets and similar electronic enclosures. OctaBDEs have been largely utilized in plastics for business equipment, whereas pentaBDEs have been principally used in foam for cushioning in upholstery.

Since PBDEs can migrate from original products into the surrounding elements, they are likely to disseminate into air, water, sediments, soil and sludge at places where they are produced, used or disposed [10] (see Table 1). The greatest concern is for particle-bound PBDEs released into the atmosphere, with significantly higher levels found in indoor air than in outdoor air. Although lower, PBDEs emissions in soil and water may be significant for human exposure as well [11]. Some congeners and their adverse health effects are summarized in Table 2.

Atmospheric dissemination of PBDEs as vapor or particles contributes to dust composition. Electronic devices treated with PBDEs represent an important source of air pollution as their ability to release PBDEs can be maintained over time. Among the decaBDE congeners, 2,3,4,5,6-Pentabromo-1-(2,3,4,5,6-pentabromophenoxy)benzene (BDE-209) has the most characterized toxicological profile: in Chinese areas of industrial production, BDE-209 has been detected in both maternal and cord blood samples, despite being a large, bulky molecule with a very low solubility in water.

In the United States, the major source of contamination is represented by dietary intake followed by indoor dust [12] and air, such as via house dust inhalation, ingestion or dermal exposure. In toddlers, who ingest more dust than adults, the indoor PBDE exposure is concerning [13]. In fact, children’s exposure to indoor dust is a greater source of PBDEs than diet [14]. In infants, breastfeeding may become an additional route of PBDE absorption [15]. Luckily, a review that reported the results from the 2014–2021 HBM4EU aligned studies, regarding levels of flame retardants in children from nine European countries, showed a reduced concentration of 2,2′,4,4′-tetrabromodiphenyl ether (BDE-47), probably due to implemented PBDE restrictions [16].

With respect to the decaBDEs, the majority of lower-brominated PBDEs are highly lipophilic organic compounds, able to accumulate and persist in lipid-rich tissues. Bio-concentration of pentaBDEs is possible from aquatic organisms to fishes. In areas close to dumpsites, landfills, transfer stations and trash recycling facilities, low levels of PBDEs have also been measured in meat and dairy products, supposedly resulting from animal exposure to past or ongoing emissions from older products. In Europe, the diet-derived ingestion of PBDEs is considered the predominant route of exposure [9].

### 2.2. Dose-Response and Time-Exposure Curves

According to a fundamental pharmaco-toxicological principle, there is a correlation between the severity of an effect and the compound/toxic exposure level (linear dose–effect relationship). This concept implies that no effect can be detected for exposures below the lowest-observed-adverse-effect level (LOAEL). However, as chemicals able to disrupt the endogenous hormonal activities, PBDEs may be characterized by a non-linear, non-monotonic dose–response (NMDR) relationship (see Figure 2) [17] and by effects observed below the LOAEL [14,15].

The NMDR behavior includes the concept of “hormesis” [18], in which low doses and high doses may exert opposite and diverging effects. For instance, on the same physiological parameter, low doses may result in a stimulatory effect, while inhibitory effects prevail at high doses [19]. As an important consequence of NMDR curves, the identification of risk exposure might not be visible in the short-term, and the identification of a causative dose–effect mechanism might be delayed or unattainable. Additionally, under long-term continuous exposure, PBDE toxicity may manifest with paradoxical effects: PBDEs might trigger the protective adaptive response of metabolic pathways aimed at elimination of chemicals in the short term but transiently enhance the accumulation of very reactive intermediate compounds able to exert toxic effects in the long term [20].

In other words, according to the time interval, the same pathway involved in the PBDEs could result in adaptive (short-term) or toxic (long-term) consequences. For long-term effects, it should also be taken into consideration that chemicals poorly metabolized and eliminated may persist in adipose tissue; the latter, acting as a storage depot, may become an internal source of continuous exposure [21].

Although not conclusive, the current evidence suggests that PBDEs may expose subjects to both epigenetic toxicity and genotoxicity. However, even if in both cases the effects become visible in the long term, epigenetic marks may undergo reversible changes, whereas mutations are irreversible.

### 2.3. Target Vulnerability

The importance of exposure to several EDs (including PBDEs) during developmental stages is confirmed by both experimental and epidemiological studies reporting an increase in disease risk later in life [22]. Developing organisms are highly vulnerable, due to the remodeling of tissues and organs and to limited defense mechanisms. Thus, either continuous or limited exposure can profoundly impact on subsequent evolution and may induce epigenetic changes somatically inheritable [23]. Such alterations are uncommonly visible at birth, making it difficult to detect the causal relationship with the damaging substances. However, even subtle changes in organ physiology may affect the proper development and adaptation, and therefore contribute to increased risk of disease onset later in life.

### 2.4. Combination Effect of Mixtures

Although decaBDEs and lower-brominated PBDEs display important differences in pharmacokinetics and toxicity profiles, toxicity data referring to individual congeners may over- or underestimate the actual health risk of PBDE mixtures. Indeed, since toxic potency of a single component may be influenced by other congeners in an additive, less-than-additive or more-than-additive way, the overall effects are difficult to predict. In addition, the nature of PBDEs to which subjects may be exposed can differ from the original PBDE source for several reasons, including changes in congener composition due to instability, partitioning and alteration in the environment, and/or individual discrepancies in preservation, biotransformation and storage in the human tissues [24].

## 3. Interactions between PBDE and the Endocrine System: Focus on Molecular Mechanisms

### 3.1. Insulin Receptors

Only a small number of studies evaluated the effect of the exposure to PBDEs on insulin receptor expression and signaling [25]. It has been demonstrated that decaDBE exposure markedly increased fasting blood glucose levels and significantly reduced mRNA levels of insulin receptor (*Insr*) and of glucose transporter type 4 (*Glut4*) in the skeletal muscle of mice fed with a normal diet (ND) but not in mice fed with a high-fat diet (HFD), already characterized by a reduction in Insr gene expression [25]. Conversely, in the epididymal fat of these mice, no differences in mRNA levels of *Insr* were found [25]. In addition, in brown adipose tissue, decaDBE reduced *Insr* mRNA levels in the HFD group and insulin receptor substrate (*Irs*)1 mRNA levels in the ND group, compared to the control group [25]. Furthermore, in ND-fed mice, exposure to decaBDE decreased glucose transporter type 2 (*Glut2*) mRNA levels in the liver, while no effect was observed in the HFD-fed mice [25].

In another study on mice, 4 weeks of exposure to 2,2′,4,4′,5,5′-Hexabromodiphenyl ether (BDE-153) was found to alter glucose and lipid metabolism; there was a nonlinear dose–response relationship between its dose and the resulting interference with expression of PPARγ, AMPKα and adipokines [26].

Krumm et al. [27] demonstrated that PBDEs, whether interacting with estrogen receptor (ER)α or peroxisome proliferator-activated receptor gamma (PPARγ), are able to increase the expression of peptide hormone receptors in the arcuate nucleus of the hypothalamus in mice, including *Insr* (6- to 8-fold in males, 2- to 3-fold in females), especially in PMOC and NPY neurons, thus affecting energy balance through the hypothalamic modulation of food intake.

Interestingly, it has also been demonstrated that BDE-47 and 2,2′,3,4,4′-Pentabromodiphenyl ether (BDE-85) enhance glucose-stimulated insulin secretion in INS-1 832/13 pancreatic β-cells by binding to the thyroid hormone receptor and activating the AKT signaling pathway [28]. This fact proves that β-cell function could be influenced by environmental pollutants, such as PBDEs. It can be suggested that chronic PBDE exposure could induce hyperinsulinemia and, in turn, insulin resistance [28].

Although these findings suggest that PDBEs may affect insulin receptor expression and signaling, details of their molecular mechanisms are not fully understood and further studies are therefore warranted in order to better elucidate these effects.

### 3.2. Thyroid Hormone Receptors

The thyroid hormone (TH) signaling pathway is highly regulated, mainly through the expression of thyroid hormone receptor (TR) isoforms, cell- and tissue-specific TH transporters [29]. Two TR genes have been described (TRα and TRβ), with varying expression patterns in both developmental and adult tissues [30].

PBDEs and their metabolites, hydroxylated PBDEs (OH-PBDEs), could have an important disrupting effect on thyroid function [31]. Specifically, PBDE sulfates, through their structural similarity with THs (Figure 3), could bind thyroxine-binding globulin (TBG) and transthyretin (TTR), two important TH transport proteins. Through site-specific fluorescence probes, it has been demonstrated in vitro that OH-PBDEs bind TH transport proteins such as TBG and TTR [32]. In this regard, OH-PBDEs are structurally similar to THs, resulting in an in vitro demonstrated stronger binding capacity to TTR than thyroxine [33].

PBDEs and OH-PBDEs could also affect the two subtypes of nuclear TR. Agonistic activity of PBDEs on transcription factor TRβ in the human thyroid follicular cell line has been demonstrated [31]. Therefore, PBDE sulfates could disrupt TH signaling through the interaction with TH transport proteins or TRs, as seen in an in vitro study [34]. 

Furthermore, PBDE may also affect TH signaling by interacting with retinoic acid (RA) signaling: in zebrafish larvae, BDE-47 was able to alter several steps of the RA pathway, such as RA metabolism and receptor binding, which may in turn alter TH signaling [35].

### 3.3. Other Receptors

PBDEs can interact with several other receptors, in particular steroid receptors, acting as transcriptional factors and modulating several steroid-dependent effects. Indeed, PBDEs can activate ERα, while they antagonize androgen receptor (AR) and progesterone receptor (PR) with a higher potency as compared to natural ligands or clinical drugs. These effects, shown in vitro, have been ascribed to the similar chemical structure between steroid hormones and several PBDEs that can directly bind steroid hormone receptors modulating their activity and, in some cases, impair binding with the natural ligands [36,37]. Most of the PBDE compounds with estrogenic activity, for example, interact with the binding cavity of ER similarly to estradiol (E2) [38]. Some of these compounds have also been shown to inhibit estradiol sulfotransferase, an enzyme which increases the sulfated estrogens’ hydrosolubility, preventing their binding with the ER [39], thereby explaining the anti-estrogenic actions of selected PBDEs [36].

In addition, PBDEs can interact with the PPAR family, since they reduce PPARγ transactivational activity, impairing the transcription of antioxidative enzymes (i.e., glutathione peroxidase, superoxide dismutase). This action is due to a reduced interaction between PPARγ and PPARγ coactivator-1 alpha (PGC1α) [40].

An antagonistic action has been demonstrated for PBDEs towards glucocorticoid receptors (GR), as well [41]. Therefore, PBDEs might profoundly disrupt steroid hormone receptor signaling.

## 4. Clinical Effects of PBDE Exposure

### 4.1. Diabetes, Insulin Resistance, Obesity and Metabolic Syndrome

The well-known association between diabetes, insulin resistance, obesity and metabolic syndrome involves both genetic susceptibility and exposure to environmental factors.

Genetic variants, from genome-wide association studies, can explain only about 10% of the phenotypic variability, suggesting that the environment might play a crucial role [42]. Pollutants such as PBDEs have been suggested to represent additional risk factors for glucose metabolism dysregulation and diabetes, even though data from the literature are not fully consistent [43]. Different study designs, limited sample sizes and methodological differences, as well as confounders (e.g., family history of diabetes, diet and exercise), might explain such inconsistent findings.

Examining data from the National Health and Nutrition Examination Survey (NHANES) 2003–2004, authors described a U-shaped relationship between diabetes and metabolic syndrome on one side and exposure to BDE-153 on the other [44]. Conversely, a positive linear association between BDE-47 serum concentration and type 2 diabetes risk was found by Zhang et al. in two Chinese rural cohorts [45]. Some potential associations between dietary exposure to PBDEs and the risk of diabetes were observed in a French prospective cohort [46]. Moreover, consistent data showed that exposure to PBDEs increases the risk of gestational diabetes mellitus (GDM) [47]. Conversely, studies conducted in North America, in Finland and in Sweden failed to demonstrate such an association [48,49,50].

Potential associations need to be further confirmed through larger epidemiological and experimental studies.

### 4.2. Thyroid Disease

Given their structural similarity to THs (see Figure 3), PBDEs could disrupt TH levels and signaling, potentially altering their effects, both during development and adult life [33,51,52,53,54].

Some experimental studies in rodents showed that PBDE exposure may lead to hypotyroxinemia [53,55,56]. Although several toxicological studies showed a possible association with thyroid dysfunction, no strong association has been demonstrated at the epidemiological level between PBDE exposure and thyroid disruption [57,58,59,60].

Makey et al. described how PBDE exposure influenced thyroid function in a longitudinal North American cohort of healthy adults: exposure to PBDEs was associated with reduced TT4 levels, while other thyroid function parameters were not influenced. This evidence suggests that PBDEs’ effect on TT4 may derive from a decreased plasma protein binding of T4 without interference on the pituitary–thyroid axis [60]. In women too, especially after menopause, exposure to PBDEs has been associated with thyroid disease, thus suggesting a role for lowered estrogen levels [61].

There is a need for further prospective studies with the aim of determining how PBDEs may influence and disrupt thyroid function in humans.

### 4.3. Pubertal Effects

Due to their xenohormonal proprieties, PBDEs may also have androgen-like and estrogen-like effects and affect puberty timing and regulation. Several observational studies were carried out to elucidate potential effects. A 10-fold increase in maternal prenatal PBDEs serum levels resulted in increased FSH, LH and testosterone serum levels in a cohort of 234 male children aged 12 years [62].

An Italian study [63] described an association between premature thelarche and PBDE serum levels, somehow confirming the results of another study which found increased PBDE serum levels in girls with idiopathic central precocious puberty [64].

Studies on menarche timing showed conflicting results. In utero and childhood, exposure to PBDEs resulted in later age at menarche in girls and earlier pubarche in males [65]. Another study confirmed the results in females; however, BMI-corrected hazard ratios attenuated the correlation [66]. This is in contrast with an older study [67] where higher serum levels of PBDEs were associated with a slightly earlier menarche onset.

Due to the sparse and conflicting results from available data, more studies are warranted to elucidate the associations between PBDEs and puberty regulation.

### 4.4. Ovarian Function and Reproductive Health

Among the possible harmful effects on human health, PBDEs may also affect the reproductive system. A recent review [47] evaluated the risk of GDM after PBDE exposure. Four studies were analyzed, evaluating six PBDE congeners and total PBDE exposure, estimating the odds ratio (OR) with a random-effects model. The authors found a positive association between PBDE exposure and the risk of GDM (OR 1.32, 95% CI = 1.15–1.53). The main contributor to this increased risk was 2,2′,4,4′,5,6′-Hexabromodiphenyl ether (PDBE-154) (OR: 1.23; 95% CI  =  1.13–1.35), and the pooled OR for total PBDE exposure was 2.21 (95% CI  =  1.55–3.16). The other congeners did not show a significant association. Such analysis suggests that exposure to PBDEs, mainly PBDE-154, could lead to GDM. Conversely, another study [68] found lower levels of PBDEs in placenta of mothers with GDM when compared to controls.

Furthermore, in a mouse model, perinatal exposure to a PBDE mixture resulted in a diabetic phenotype in the offspring [69].

Regarding male reproductive health, a recent review [6] found that PBDE exposure might impair semen quality, but with limited evidence, and that different congeners might elicit different LH and testosterone responses, with conflicting results. A recent study showed that prenatal exposure of mice to 2,2′,4,4′,5-Pentabromodiphenyl ether (PBDE-99) caused spermatogenic injuries, due to several mechanisms such as increased levels of reactive oxygen species and dysfunction of autophagy [70].

Finally, the exposure of pregnant women to PBDEs may increase telomer length in newborns, which may subsequently increase cancer risk in adulthood [71]. Higher placental PBDE levels were also associated with lower head circumference and Apgar score at 1 min [72].

## 5. Cancer and PBDEs

### 5.1. Thyroid Cancer

Thyroid malignancies are the most common endocrine cancers, and their incidence rate has increased in many countries in recent years [73]. Recent studies have highlighted the role of EDs in thyroid homeostasis and risk of thyroid cancer. For instance, a multicenter, cross-sectional study pointed out that the exposure to Bisphenol E, Bisphenol AF and Bis(2-ethylhexyl) phthalate was related to a significantly higher risk of differentiated thyroid cancer (DTC) [74]. Some studies have shown an effect of PBDEs on TH levels in patients operated on for thyroid cancer [75], but only a few of them were conducted in vitro to estimate the risk of thyroid cancer associated with PBDEs [31,32]. 

A recent in vitro and in vivo mouse model study demonstrated that long-term exposure (defined as more than 27 days) to environmental concentrations of decaBDE (BDE-209) increased the proliferation of normal human follicular epithelial cell line and papillary thyroid carcinoma (PTC)-derived cell lines [76]. Specifically, as shown in another in vitro and in vivo study, BDE-209 stimulates cell proliferation during both the S and the G2/M phases of the cell cycle, acting as a competitive inhibitor of TRß expression and function [76]. Furthermore, an in vitro study on human thyroid follicular cell lines indicated that cells exposed to PBDEs showed DNA single-stranded and double-stranded breaks [77]. A case–control study involving 308 thyroid cancer patients and 308 controls showed that exposure to heavy metals and to some PBDEs had a positive effect on thyroid cancer risk [78]. Conversely, a case–control study investigating the association between PBDE exposure and PTC risk in 250 females with thyroid cancer and 250 age-matched controls did not find an increased risk of PTC related to the exposure [79]. 

Overall, these studies suggest an important role of EDs, specifically of PBDEs, in thyroid tumorigenesis. However, further investigation is needed to better clarify the pathogenesis and clinical impact.

### 5.2. Breast Cancer

The increase in breast cancer incidence in recent decades, associated with higher prevalence of this malignancy in industrial urban areas, has led to speculation that environmental pollutants, such as PBDEs, might be involved in the etiology of breast cancer. Three small case–control studies did not find a significant association between the risk of breast cancer and PBDE levels [80,81,82]. To evaluate the risk of invasive breast cancer associated with serum PBDE levels, another case–control study was conducted among 1838 women [83]: 902 cases with invasive breast cancer and 936 matched controls were interviewed from 2011 to 2015, collecting blood samples from cases at an average time of 35 months after breast cancer diagnosis. The results did not provide any evidence of an association between serum levels of BDE-47, 2,2′,4,4′,6-Pentabromodiphenyl ether (BDE-100) and BDE-153 with breast cancer risk. However, this study had some limitations. First, PBDE measurements may have not represented chronic or early pre-diagnostic exposure. Secondly, there was a lack of information about genetic polymorphisms and other factors potentially influencing endogenous estrogen levels. An association between concentrations of PBDEs in the adipose tissue and breast cancer risk was observed in a case–control study in women in Chaoshan, China [84]. Studies in vitro showed that PBDEs might have estrogenic effects and could stimulate cell proliferation [85,86,87,88].

Further experimental and epidemiological studies are required to confirm this potential association and to understand the mechanism involved.

### 5.3. Liver Cancer

PBDEs may be able to affect liver function: Dunnick et al. assessed mice liver transcriptomic patterns to characterize and compare the toxicity of PBDE-47 to that of a PBDE mixture to help predict the likelihood of triggering carcinogenesis and long-term toxicity [89]. Liver transcript data were used for preliminary benchmark dose risk analysis as was carried out for other chemicals inducing liver toxicity and cancer. The results showed that DE-71, a mixture of PBDEs, can cause liver toxicity, while PBDE-47 can induce centrilobular hypertrophy and fatty changes in liver.

DE-71 is a PBDE mixture and a liver carcinogen in rats and mice, according to the National Institute of Environmental Health Science/National Toxicology Program. Molecular events downstream of TBX3 (a protein-coding gene) suggest a possible mechanism by which DE-71 could influence hepatocellular carcinogenesis.

Shimbo et al. used liver carcinoma tissue from mice exposed to DE-71 to assess the hypothesis that DNA methylation, an epigenetic alteration consistent with tumor development, might derive from long-term DE-71 exposure. Indeed, this exposure may induce tumor development by epigenetic programs that facilitate expansion of progenitor cell populations [90].

PBDE carcinogenic potential might also be related to oxidative damage, to hormone homeostasis disruption and to molecular and epigenetic alterations in target tissues, as shown in mice [91]. Apparently, no published data on the carcinogenic effect of PBDEs on the liver in human case series are available. Further research is needed to compare the PBDE toxic effects in mice and humans. 

### 5.4. Other Cancers

In addition to previously reported data, several studies have suggested that PBDEs can enable critical cancer hallmark mechanisms, affecting genomic integrity and promoting inflammatory signaling pathways [92,93].

It has also been hypothesized that these environmental chemicals function as “immune disruptors” at low doses and in concert with other factors (low-dose mixture hypothesis) [94,95], thus potentially inducing molecular and epigenetic modifications that contribute to carcinogenesis in target tissues.

Although a growing number of studies have investigated the relationship between PBDEs and cancer development, limited evidence exists to elucidate carcinogenic effects of most PBDEs both in animal and in human case–control epidemiological studies, except for decaBDE, for which the U.S. Environmental Protection Agency indicated “suggestive evidence of carcinogenic potential” in 2008. Recently, the PBDEs were included in the high-priority list of agents not previously evaluated by the International Agency for Research on Cancer (IARC) Monographs [96].

A further field of investigation is the correlation between the higher environmental PBDE concentration during e-waste recycling processes and the increasing incidence of cancer (such as liver and lung cancer) reported around the e-waste dismantling areas.

In a study conducted in China [97] BDE-47 and 2,4,4′-Tribromodiphenyl ether (BDE-28) were the main congeners among the PBDE family measured in tissue samples (kidney, liver and lung) of surgical cancer patients living around these sites. Moreover, BDE-209 was identified in more than 70% of the samples. 

Several preclinical studies have evaluated the effects of specific PBDE congeners both in cancer development and in the regulation of cancer cell growth. In vitro exposure to BDE-47, 99 and 209 could induce cancerogenesis in airway epithelial cells with the activation of EZH2 methyl transferase, the expression of H3K27me3 and k-RAS proteins, and ERk1/2 phosphorylation [98].

A recent in vitro study presented new insights into colorectal cancer pathogenesis, highlighting the effects of BDE-99 exposure (one of the PBDE congeners most widespread in the environment and in human tissues) on colorectal cancer cells, 28 (EMT) through the PI3K/AKT/Snail signaling pathway [99].

This evidence suggests the intriguing role of some PBDEs in cancer pathogenesis. Moreover, further data are needed to better define their potential activity in cancerogenesis and identify specific molecular mechanisms and key signaling pathways.

### 5.5. PBDE and Inflammation

PBDE can also affect immune response and function at various levels. An Italian microarray, in vitro and in silico study [100] observed that BDE-47 modulated the expression of many intracellular miRNAs, altering the innate immune response and macrophage function. This modulation might also affect cancer risk: a subsequent study [101] showed that macrophages exposed to BDE-47 secrete extracellular vesicles that alter the function of lung epithelium, thereby probably increasing risk for further lung diseases and cancer.

## 6. Regulations and Efforts on Preventing Exposure to PBDE

In the IARC monograph on PBDEs [102], these substances are considered “probably carcinogenic to humans”, sharing several similarities with polychlorinated biphenyls; in their review, the authors reach conclusions similar to ours. They state that some congeners are involved in two-stage hepatocarcinogenesis, that they are ligands for cellular and nuclear receptors and that they can induce microscopic changes in rodent liver and thyroid, acute toxicity in liver and thymus, reduce immunocompetence in animals and humans, increase hepatic metabolism of estrogens in vitro and increase the odds of male birth in women.

Due to the health concern for human exposure to PBDE, the United States Environmental Protection Agency (EPA) made an action plan [103] to reduce the risk of exposure to these substances: a proposal for a significant new use rule (SNUR) that requires notice to EPA prior to manufacturing or importing articles with commercial (c-) c-pentaBDE or c-octaBDE; “support and encourage the voluntary phase-out of manufacture and import of c-decaBDE”; “initiate rulemaking to propose a simultaneous SNUR and test rule for c-decaBDE”. They also state a no-observed-adverse-effect level (NOAEL) for PBDE of 1.77 mg/kg-day for induction of hepatic enzymes [104].

Minimal risk levels (MRL) set by the U.S. Agency for Toxic Substances and Disease Registry [9] are as follows: regarding inhalation exposure, for lower-brominated BDEs “an MRL of 0.006 mg/m^3^ has been derived for intermediate-duration inhalation exposure (15–364 days)”; for decaBDE, there were insufficient data on inhalation exposure to set an MRL.

Regarding oral exposure, for lower-brominated BDE, “an MRL of 0.00006 mg/kg/day has been derived for acute-duration oral exposure (14 days or less)” and “an MRL of 0.000003 mg/kg/day has been derived for intermediate-duration oral exposure (15–364 days)”; for decaBDE, “an MRL of 0.01 mg/kg/day has been derived for acute-duration oral exposure (14 days or less)” and “an MRL of 0.0002 mg/kg/day has been derived for intermediate-duration oral exposure (15–364 days)”.

## 7. Conclusions

Diabetes, obesity, metabolic diseases and cancer are widespread conditions all over the world and major threats to public health, with an enormous social impact both in terms of direct medical costs and in terms of lost productivity. They have traditionally been regarded as multifactorial disorders, with lifestyle influences as well as strong genetic susceptibility and environmental components. Moreover, all of these conditions share well-established risk factors, such as familiarity, aging, insulin resistance, hyperinsulinemia, chronic inflammation and abnormal cytokine secretion. Indeed, the simultaneous occurrence of these diseases in the same patient is much more common than we would expect if this was to occur by chance [105].

Nevertheless, many aspects of the relationship between these common diseases are still unknown, and there is still a lot of room in this area for basic research. Lately, in addition to conventional risk factors, environmental factors have been suggested to make a significant contribution. Specifically, exposure to environmental chemicals, persistent organic pollutants and EDs have been associated with both type 2 diabetes (and related metabolic conditions) and some cancers [49]. Furthermore, PBDEs can interact with several hormone receptors, such as steroid and thyroid hormone receptors, acting as transcriptional factors, modulating several endocrine effects and profoundly disrupting hormone receptors’ signaling, with a relevant impact on overall metabolism and on reproductive health, as summarized in Table 3. 

A deeper understanding of the complex interplay between both developmental and adult chronic exposure to EDs, on one side, and obesity, diabetes, related metabolic disturbances and cancer, on the other side, is a high-priority research goal and responds to the growing need to identify environmental modifiable risk factors that can help guide public health interventions, in order to reduce the burden of these major social threats [43]. Basic research and epidemiological efforts can strongly contribute to identify relevant biological pathways and shared mechanisms for the observed associations, as well as potential solutions that, if globally implemented, could reduce disease incidence, with its associated health and economic burden. Similarly, scientific societies can play a pivotal role in presenting to all the stakeholders, starting from public health authorities, the potential health risks of environmental pollutants such as PBDEs and in guiding public health interventions. Finally, interventions for environmental improvement are urgently needed to reduce/eliminate the continuing burden of environmental contamination, particularly near electronic waste recycling sites.

## Figures and Tables

**Figure 1 cancers-15-04237-f001:**
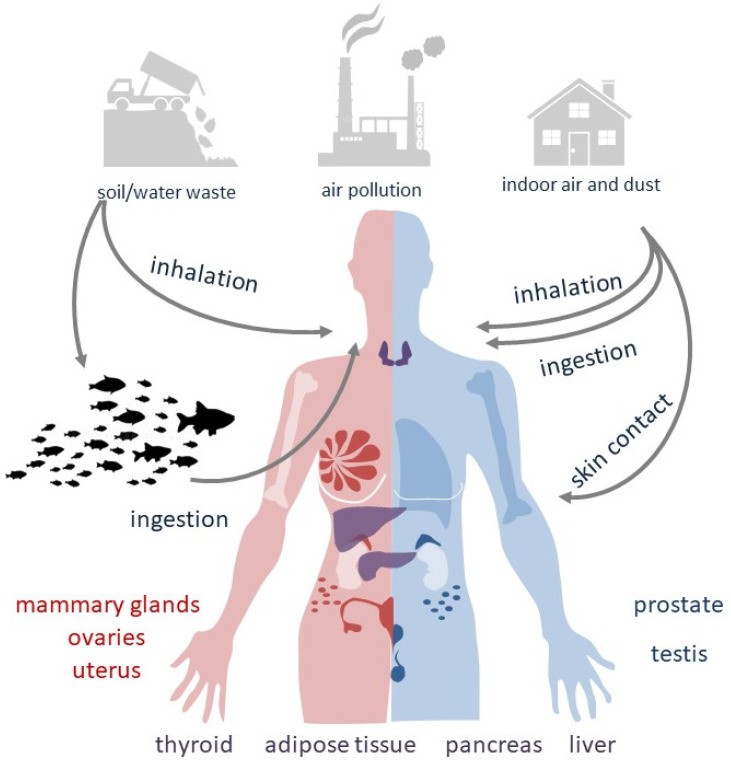
Routes of exposure to PBDEs.

**Figure 2 cancers-15-04237-f002:**
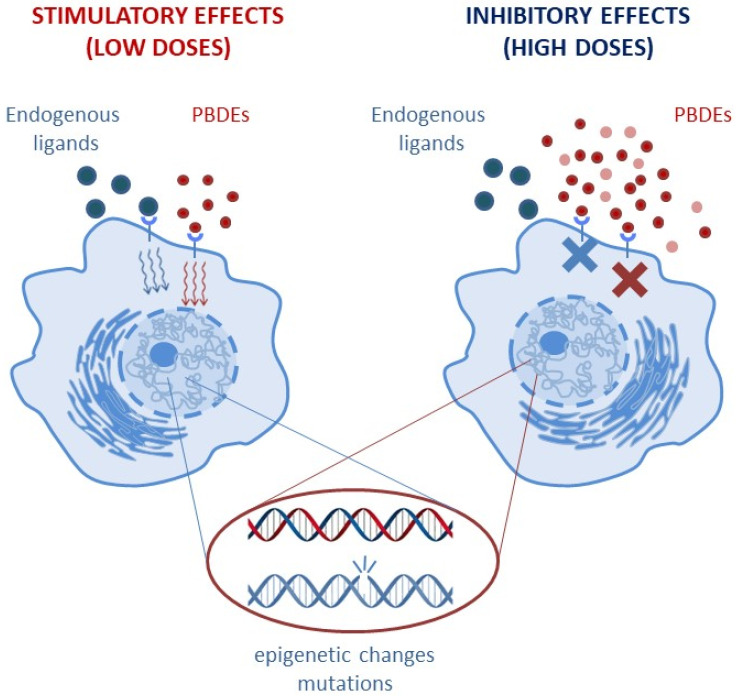
PBDEs may be characterized by a non-monotonic dose–response (NMDR) relationship, with low doses triggering stimulatory effects and high doses enhancing inhibitory effects on the same physiological parameter.

**Figure 3 cancers-15-04237-f003:**
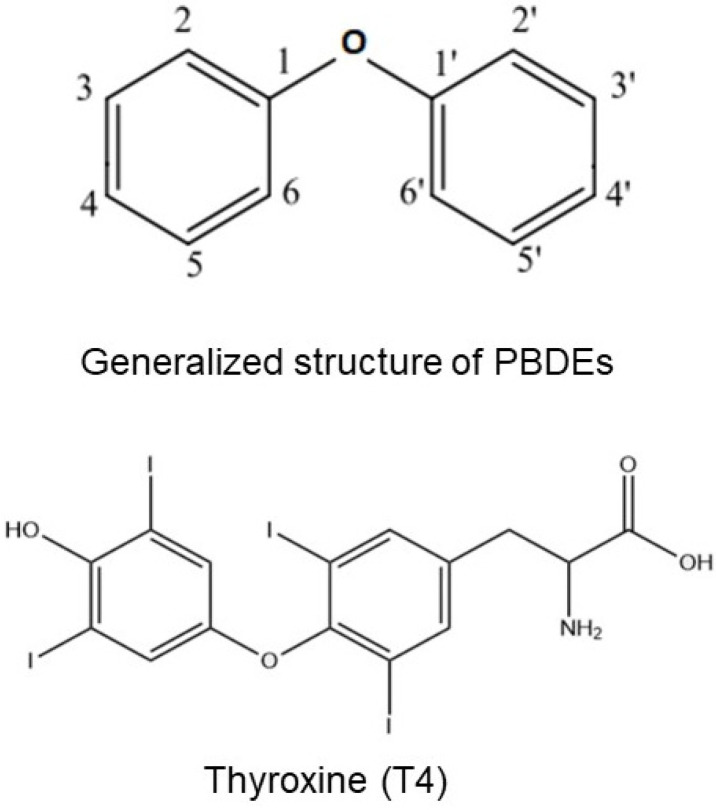
Structural similarities between PBDEs and T4.

**Table 2 cancers-15-04237-t002:** Some PBDE congeners and their adverse effects. GDM: gestational diabetes mellitus.

Congener	Effects
2,3,4,5,6-Pentabromo-1-(2,3,4,5,6-pentabromophenoxy)benzene (BDE-209)	Long-term exposure increased proliferation of normal human thyroid follicular epithelial cell line, papillary thyroid carcinoma (PTC)-derived cell, in vitro and in mice. Could induce cancerogenesis in airway epithelial cells. Most toxicologically characterized PBDE.
2,2′,4,4′-tetrabromodiphenyl ether (BDE-47)	Alters retinoic acid pathway in zebrafish larvae. Linear association between BDE-47 serum concentration and type 2 diabetes risk. Centrilobular hypertrophy and fatty changes in liver. Could induce cancerogenesis in airway epithelial cells.
2,2′,4,4′,5,5′-Hexabromodiphenyl ether (BDE-153)	Alteration of glucose and lipid metabolism in mice. U-shaped relationship between diabetes and metabolic syndrome on one side and exposure to BDE-153 on the other.
2,2′,3,4,4′-Pentabromodiphenyl ether (BDE-85)	Enhances glucose-stimulated insulin secretion in INS-1 832/13 pancreatic β-cells.
2,2′,4,4′,5,6′-Hexabromodiphenyl ether (BDE-154)	Increase in GDM risk.
2,2′,4,4′,5-Pentabromodiphenyl ether (BDE-99)	Spermatogenic injuries in prenatally exposed rats. Could induce cancerogenesis in airway epithelial cells. Activation of epithelial–mesenchymal transition in colorectal cancer cells.
DE-71 (mixture)	Liver toxicity.

**Table 3 cancers-15-04237-t003:** Summary of toxicity studies analyzed in this review.

	In Vitro	Animal Models	Human
Insulin receptors	Enhance glucose-stimulated insulin secretion in INS-1 832/13 pancreatic β-cells [28].	Increase in fasting blood glucose and reduced mRNA levels of Insr and Glut4 in mice fed with a normal diet [25].Alteration of glucose and lipid metabolism in mice [26].Increased expression of peptide hormone receptors in mice hypothalamus, affecting energy balance [27].	
Thyroid hormone receptors	Agonistic activity of PBDEs on transcription factor TRβ in the human thyroid follicular cell line [31].OH-PBDEs bind THs transport proteins such as TBG and TTR [32].PBDE stronger binding capacity to TTR than thyroxine [33].PBDE sulfates could disrupt THs signaling through the interaction with THs transport proteins or TRs [34].	Alteration of retinoic acid pathway in zebrafish larvae [35].	
Other receptors	Directly bind steroid hormone receptors [36,37].Interaction with estrogen receptors [38].Inhibition of estradiol sulfotransferase [39].Reduced PPARγ transactivational activity [40].Antagonistic action towards glucocorticoid receptors [41].		
Diabetes, insulin resistance, obesity and metabolic syndrome			U-shaped relationship between diabetes and metabolic syndrome and PBDE-153 [44].Positive linear association between PBDE47 serum concentration and type 2 diabetes risk [45].Potential association between dietary PBDE and risk of diabetes [46].Increased risk of gestational diabetes mellitus (GDM) [47].No association between PBDE exposure and diabetes risk [48,49,50].
Thyroid disease		Hypothyroxinemia in rodents [53,55,56].	No strong association between PBDE exposure and thyroid dysfunction [57,58,59,60].Reduced TT4 levels [60].Thyroid disease in women [61].
Pubertal effects, ovarian function and reproductive health		Diabetic offspring in mice [69].Spermatogenic injuries in prenatally exposed mice [70].	Increased FSH, LH and testosterone in children [62].Premature thelarche [63]Increased levels of PBDE in girls with precocious puberty [64].Prenatal and childhood exposure to PBDE resulted in later age at menarche in females [65,66] and earlier pubarche in males [65].Earlier menarche onset [67].Increased risk of GDM [47].Impaired semen quality [6].Increase telomer length in newborns [71].Lower head circumference and Apgar at 1 min [72].
Thyroid cancer	DNA single-stranded and double-stranded breaks [77].	Increased proliferation of normal human follicular epithelial cell line and papillary thyroid carcinoma (PTC)-derived cell lines (both in vivo and in vitro) [76].Cell proliferation stimulus (both in vivo and in vitro) [76].	Effect on thyroid hormone levels in patients operated for thyroid cancer [75].Case–control study showed increased thyroid cancer risk [78].Case–control study did not show increased papillary thyroid cancer risk [79].
Breast cancer	Possible estrogenic effect and therefore proliferation stimulus [85,86,87,88].		No significant association with breast cancer [80,81,82,83].Increased breast cancer risk [84].
Liver cancer		Liver toxicity in mice [89].Potential causality for PBDE exposure in liver carcinomas from mice [90].Liver cancer in mice [91].	
Other cancers	Cancerogenesis in airway epithelial cells [98].Activation of epithelial–mesenchymal transition (EMT) in colorectal cancer cells [99].		PBDE found in tissue samples from surgical cancer patients [97].
Inflammation	Altered innate immune response [100,101].

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
