# Peer review of "Polybrominated Diphenyl Ethers (PBDEs) and Human Health: Effects on Metabolism, Diabetes and Cancer"

_cancers, 2023, doi:10.3390/cancers15174237_

Round 1

Reviewer 1 Report

Review of Cancers Paper Submission:

 “Polybrominated Diphenyl Ethers (PBDE) and Human Health: Effects on Metabolism, Diabetes and Cancer. An Italian Association of Medical Oncology (AIOM)/ Italian Association of Medical Diabetologists (AMD)/ Italian Society of Diabetology (SID), Italian Society of Endocrinology (SIE)/ Italian Society of Pharmacology (SIF) Multidisciplinary Narrative Review.”

Overall Review:  In this paper a group of Italian Societies and Scientists review the metabolism and endocrine disruption and other toxic effects of polybrominated diphenyl ethers (PBDEs).  This review is a good summary of the toxicity studies performed for the PBDEs.

Comments and suggestions:

1.     The paper reviewed data from humans and animals.  However, the text of the paper as written does not always clarify where the data came from.  Please review the text of the paper to make sure the medium for the finding is clear (i.e., animal/in vitro/or human study.  For example, it should be clear that the following study was done in mice.  

·       “It has been demonstrated that decaDBE  exposure markedly increased fasting blood glucose levels and significantly reduced mRNA levels of insulin receptor (Insr) and of glucose transporter type 4 (Glut4) in the skeletal muscle of mice fed with a normal diet (ND), but not in mice fed with a high-fat diet (HFD), already characterized by a reduction in Insr gene expression [25] (should clarify in the text of the paper that these studies were done in mice).”

2.     Include a table summarizing the toxicities studies (reviewed in this paper) performed in animals, in vitro, or findings from human studies.

3.     Include information describing the sensitivity of humans vs. animals to the toxic effects of PBDEs.   Some studies provide data that the PBDEs cause thyroid hormone disruption at much lower doses than in animals.  See two papers on this topic 1,2 .

4.     Add a section to the paper on regulations/guidelines/information for preventing exposures to PBDEs.

·       Include information on minimal risk levels established by ATSDR.

·       Include information on EPA’s actions on PBDEs and reference dose levels.

https://www.epa.gov/assessing-and-managing-chemicals-under-tsca/polybrominated-diphenyl-ethers-pbdes

https://iris.epa.gov/ChemicalLanding/&substance_nmbr=184

·       Include a summary of IARC’s review of PBDEs

1.         Cowell WJ, Sjödin A, Jones R, et al. Pre- and Postnatal Polybrominated Diphenyl Ether Concentrations in Relation to Thyroid Parameters Measured During Early Childhood. Thyroid. 2019;29(5):631-641.

2.         Makey CM, McClean MD, Braverman LE, et al. Polybrominated Diphenyl Ether Exposure and Thyroid Function Tests in North American Adults. Environmental health perspectives. 2016;124(4):420-425.

Author Response

Review of Cancers Paper Submission:

 “Polybrominated Diphenyl Ethers (PBDE) and Human Health: Effects on Metabolism, Diabetes and Cancer. An Italian Association of Medical Oncology (AIOM)/ Italian Association of Medical Diabetologists (AMD)/ Italian Society of Diabetology (SID), Italian Society of Endocrinology (SIE)/ Italian Society of Pharmacology (SIF) Multidisciplinary Narrative Review.”

Overall Review:  In this paper a group of Italian Societies and Scientists review the metabolism and endocrine disruption and other toxic effects of polybrominated diphenyl ethers (PBDEs).  This review is a good summary of the toxicity studies performed for the PBDEs.

Thank you.

Comments and suggestions:

  1. The paper reviewed data from humans and animals. However, the text of the paper as written does not always clarify where the data came from.  Please review the text of the paper to make sure the medium for the finding is clear (i.e., animal/in vitro/or human study.  For example, it should be clear that the following study was done in mice. 
  • “It has been demonstrated that decaDBE exposure markedly increased fasting blood glucose levels and significantly reduced mRNA levels of insulin receptor (Insr) and of glucose transporter type 4 (Glut4) in the skeletal muscle of mice fed with a normal diet (ND), but not in mice fed with a high-fat diet (HFD), already characterized by a reduction in Insr gene expression [25] (should clarify in the text of the paper that these studies were done in mice).”

Thank you for the suggestion, we clarified for each study if performed in vitro, in animal models or in humans according to your request.  

  1. Include a table summarizing the toxicities studies (reviewed in this paper) performed in animals, in vitro, or findings from human studies.

Thanks for the suggestion, we added a table summarizing the studies reviewed and where the reported data came from.

  1. Include information describing the sensitivity of humans vs. animals to the toxic effects of PBDEs. Some studies provide data that the PBDEs cause thyroid hormone disruption at much lower doses than in animals.  See two papers on this topic 1,2 .

We are struggling to answer satisfactorily on this point. To date, we have not been able to find conclusive evidence that humans are significantly more sensitive to the toxic effects of PBDEs than animals. We have carefully read the two articles kindly indicated by the Reviewer, but -in these papers- we did not find conclusive information on this issue. Nevertheless, the reference by Makey et al. is cited in our review, since it is highly pertinent and relevant to our research. Above all, we believe that addressing this issue in detail is not fundamental to the purpose of our narrative review and we hope that the Reviewer will be able to judge our attitude favorably.

  1. Add a section to the paper on regulations/guidelines/information for preventing exposures to PBDEs.
  • Include information on minimal risk levels established by ATSDR.

  • Include information on EPA’s actions on PBDEs and reference dose levels.

https://www.epa.gov/assessing-and-managing-chemicals-under-tsca/polybrominated-diphenyl-ethers-pbdes

https://iris.epa.gov/ChemicalLanding/&substance_nmbr=184

  • Include a summary of IARC’s review of PBDEs

Thanks for your suggestions and for the extremely useful sources; we included a new paragraph regarding prevention of exposure to PBDE (“6. Regulations and efforts on preventing exposure to PBDE”) and we cited the suggested.

1. Cowell WJ, Sjödin A, Jones R, et al. Pre- and Postnatal Polybrominated Diphenyl Ether Concentrations in Relation to Thyroid Parameters Measured During Early Childhood. Thyroid. 2019;29(5):631-641.

2. Makey CM, McClean MD, Braverman LE, et al. Polybrominated Diphenyl Ether Exposure and Thyroid Function Tests in North American Adults. Environmental health perspectives. 2016;124(4):420-425.

Reviewer 2 Report

MANUSCRIPT: 2549993

TITLE: Polybrominated Diphenyl Ethers (PBDE) And Human Health: Effects on Metabolism, Diabetes and Cancer. An Italian Association of Medical Oncology (AIOM)/ Italian Association of Medical Diabetologists (AMD)/ Italian Society of Diabetology (SID), Italian Society of Endocrinology (SIE)/ Italian Society of Pharmacology (SIF) Multidisciplinary Narrative Review

The manuscript 2549993Polybrominated Diphenyl Ethers (PBDE) And Human Health: Effects on Metabolism, Diabetes and Cancer. An Italian Association of Medical Oncology (AIOM)/ Italian Association of Medical Diabetologists (AMD)/ Italian Society of Diabetology (SID), Italian Society of Endocrinology (SIE)/ Italian Society of Pharmacology (SIF) Multidisciplinary Narrative Review”, presents a review of the literature in order to associate PDBEs with effects on human health.

The manuscript presented is well structured.

The review is clearly written, well systematized and comprehensive for the topic, and literature cited is adequate and most of the papers cited are from the last five years.

Similar reviews are not known and it is of interest to the scientific community.

The conclusions are consistent and in accordance with the quotes listed.

However, some questions remain to be clarified and solved and the manuscript in the current form must be revised in minor several points as follows comments:

 1. Title – The title is too long, a shorter title such as “Polybrominated Diphenyl Ethers (PBDE) And Human Health: Effects on Metabolism, Diabetes and Cancer” is recommended.

2. The manuscript is written in a general way about the effects of PDBEs in general, it is recommended to specify where possible which isomers are involved and to make a final table including the chemical names and what the effect is. of each of the isomers on human health as well as presenting data on the relative toxicity of each of the isomers and identifying the PBDE with the greatest toxic potential similar to other pollutants such as PCBs and dioxins.

3. Please, when you present PDBEs with an abbreviation for the first time, you must previously indicate its chemical name, example instead of PDBE-47 you must write 2,2'4,4'-Tetrabromodiphenyl Ether (PBDE-47) and do the same for others that it presents in the manuscript.

4. Please complete the manuscript to include in the health effects of PDBEs a subchapter on the effects of PDBEs on the anti-inflammatory response. Please see the following reference and research related ones.

Longo V, Longo A, Adamo G, Fiannaca A, Picciotto S, La Paglia L, Romancino D, La Rosa M, Urso A, Cibella F, Bongiovanni A, Colombo P. 2,2'4,4'-Tetrabromodiphenyl Ether (PBDE-47) Modulates the Intracellular miRNA Profile, sEV Biogenesis and Their miRNA Cargo Exacerbating the LPS-Induced Pro-Inflammatory Response in THP-1 Macrophages. Front Immunol. 2021 May 7;12:664534. doi: 10.3389/fimmu.2021.664534. PMID: 34025666; PMCID: PMC8138315.

Author Response

The manuscript 2549993 “Polybrominated Diphenyl Ethers (PBDE) And Human Health: Effects on Metabolism, Diabetes and Cancer. An Italian Association of Medical Oncology (AIOM)/ Italian Association of Medical Diabetologists (AMD)/ Italian Society of Diabetology (SID), Italian Society of Endocrinology (SIE)/ Italian Society of Pharmacology (SIF) Multidisciplinary Narrative Review”, presents a review of the literature in order to associate PDBEs with effects on human health.

The manuscript presented is well structured.

Thank you.

The review is clearly written, well systematized and comprehensive for the topic, and literature cited is adequate and most of the papers cited are from the last five years.

Thank you, again.

Similar reviews are not known and it is of interest to the scientific community. Thanks a lot.

The conclusions are consistent and in accordance with the quotes listed.

Thank you.

However, some questions remain to be clarified and solved and the manuscript in the current form must be revised in minor several points as follows comments:

  1. Title – The title is too long, a shorter title such as “Polybrominated Diphenyl Ethers (PBDE) And Human Health: Effects on Metabolism, Diabetes and Cancer” is recommended.

Thank you for your suggestion. We agree that a shorter title might be better and changed the title as advised.

  1. The manuscript is written in a general way about the effects of PDBEs in general, it is recommended to specify where possible which isomers are involved and to make a final table including the chemical names and what the effect is. of each of the isomers on human health as well as presenting data on the relative toxicity of each of the isomers and identifying the PBDE with the greatest toxic potential similar to other pollutants such as PCBs and dioxins.

Thanks for the suggestions, we added a table including the adverse effects of each PBDE. As of now, available data makes identifying a single PDE as the most toxic difficult, however we included the notion that BDE-209 is the most toxicologically characterized.

  1. Please, when you present PDBEs with an abbreviation for the first time, you must previously indicate its chemical name, example instead of PDBE-47 you must write 2,2'4,4'-Tetrabromodiphenyl Ether (PBDE-47) and do the same for others that it presents in the manuscript.

Thank you for the suggestion, we indicated their chemical names as advised.

  1. Please complete the manuscript to include in the health effects of PDBEs a subchapter on the effects of PDBEs on the anti-inflammatory response. Please see the following reference and research related ones.

Longo V, Longo A, Adamo G, Fiannaca A, Picciotto S, La Paglia L, Romancino D, La Rosa M, Urso A, Cibella F, Bongiovanni A, Colombo P. 2,2'4,4'-Tetrabromodiphenyl Ether (PBDE-47) Modulates the Intracellular miRNA Profile, sEV Biogenesis and Their miRNA Cargo Exacerbating the LPS-Induced Pro-Inflammatory Response in THP-1 Macrophages. Front Immunol. 2021 May 7;12:664534. doi: 10.3389/fimmu.2021.664534. PMID: 34025666; PMCID: PMC8138315.

Thank you for your suggestion, we included a subchapter on PBDE and anti-inflammatory response.

Round 2

Reviewer 2 Report

MANUSCRIPT: 2549993_V2

TITLE: Polybrominated Diphenyl Ethers (PBDE) And Human Health: Effects on Metabolism, Diabetes and Cancer

In the revised 2549993_V2Polybrominated Diphenyl Ethers (PBDE) And Human Health: Effects on Metabolism, Diabetes and Cancer”, the authors present the new manuscript reformulated according to almost all the reviewers´ recommendations.

I congratulate the authors for their meritorious work and the improvement effort developed, having accepted the reviewers' recommendations.

However, the manuscript should be improved in its presentation, namely:

1- Tables 2 and 3 must be previously mentioned in the text of the manuscript and must not be placed in supplementary material.

2- Include tables 2 and 3 in the manuscript with the same formatting as in table 1 so that all tables are presented with the same formatting.

Author Response

Alina Talpalaru MD

Assistant Editor

Cancers

Alessandria, August 17th, 2023

Dear Drs. Talpalaru,

We are glad to resubmit our invited narrative review now entitled “Polybrominated Diphenyl Ethers (PBDE) and Human Health: Effects on Metabolism, Diabetes and Cancer”, on behalf of a multidisciplinary panel from five Italian scientific societies (AIOM, AMD, SID, SIE, and SIF), to be evaluated for publication in Cancers.

We have carefully considered the Referees’ requests and modified the manuscript accordingly: any change to the manuscript is highlighted. Please find below, in the text, the details of the revisions to the manuscript and our responses to the referees’ comments point by point, as requested.

Kind regards,

Marco Gallo, MD

(Corresponding author)

Reviewer 2

In the revised 2549993_V2 “Polybrominated Diphenyl Ethers (PBDE) And Human Health: Effects on Metabolism, Diabetes and Cancer”, the authors present the new manuscript reformulated according to almost all the reviewers´ recommendations.

I congratulate the authors for their meritorious work and the improvement effort developed, having accepted the reviewers' recommendations. Thank you.

However, the manuscript should be improved in its presentation, namely:

1- Tables 2 and 3 must be previously mentioned in the text of the manuscript and must not be placed in supplementary material.

You are right, sorry for the inconvenience. In the new version of the manuscript, we mention Table 2 and Table 3 at line 119 and line 524, respectively. We did not intend to add the two new tables as supplementary material. We simply thought that the layout with the two tables in the text would be done automatically.

2- Include tables 2 and 3 in the manuscript with the same formatting as in table 1 so that all tables are presented with the same formatting.

The three tables are now included in the manuscript with the same formatting. We kindly ask the Assistant Editor to paginate the manuscript and the three table according to the journal style.